# Solid Lipid Nanoparticles Loaded with Glucocorticoids Protect Auditory Cells from Cisplatin-Induced Ototoxicity

**DOI:** 10.3390/jcm8091464

**Published:** 2019-09-14

**Authors:** Blanca Cervantes, Lide Arana, Silvia Murillo-Cuesta, Marina Bruno, Itziar Alkorta, Isabel Varela-Nieto

**Affiliations:** 1Institute for Biomedical Research “Alberto Sols” (IIBM), Spanish National Research Council-Autonomous University of Madrid (CSIC-UAM), 28029 Madrid, Spain; smurillo@iib.uam.es (S.M.-C.); marina.bruno@student.unisi.it (M.B.); 2Centre for Biomedical Network Research on Rare Diseases (CIBERER, CIBER), Institute of Health Carlos III (ISCIII), 28029 Madrid, Spain; 3Department of Biochemistry and Molecular Biology, University of the Basque Country (UPV/EHU), Barrio Sarriena S/N, 48080 Leioa, Spain; l_arana@hotmail.com (L.A.); itzi.alkorta@ehu.es (I.A.); 4Hospital La Paz Institute for Health Research (IdiPAZ), 28029 Madrid, Spain; 5Department of Experimental and Clinical Biomedical Sciences “Mario Serio”, University of Florence, 50134 Firenze, Italy; 6Instituto Biofisika (CSIC, UPV/EHU), Barrio Sarriena S/N 48940, 48080 Leioa, Spain

**Keywords:** nanocarriers, dexamethasone, hydrocortisone, otic protection, stearic acid-based solid lipid nanoparticle (SLN)

## Abstract

Cisplatin is a chemotherapeutic agent that causes the irreversible death of auditory sensory cells, leading to hearing loss. Local administration of cytoprotective drugs is a potentially better option co-therapy for cisplatin, but there are strong limitations due to the difficulty of accessing the inner ear. The use of nanocarriers for the efficient delivery of drugs to auditory cells is a novel approach for this problem. Solid lipid nanoparticles (SLNs) are biodegradable and biocompatible nanocarriers with low solubility in aqueous media. We show here that stearic acid-based SLNs have the adequate particle size, polydispersity index and ζ-potential, to be considered optimal nanocarriers for drug delivery. Stearic acid-based SLNs were loaded with the fluorescent probe rhodamine to show that they are efficiently incorporated by auditory HEI-OC1 (House Ear Institute-Organ of Corti 1) cells. SLNs were not ototoxic over a wide dose range. Glucocorticoids are used to decrease cisplatin-induced ototoxicity. Therefore, to test SLNs’ drug delivery efficiency, dexamethasone and hydrocortisone were tested either alone or loaded into SLNs and tested in a cisplatin-induced ototoxicity in vitro assay. Our results indicate that the encapsulation in SLNs increases the protective effect of low doses of hydrocortisone and lengthens the survival of HEI-OC1 cells treated with cisplatin.

## 1. Introduction

The use of cisplatin as a chemotherapeutic drug has important limitations due to its side effects, mainly ototoxicity [1]. This is of special relevance in pediatric cancer patients because of the impact of early hearing loss on learning and communication [2,3]. Cisplatin binds DNA bases, affecting its repair mechanisms, leading to cell apoptosis through the sum of mechanisms in which oxidative stress plays an essential role [1]. Cisplatin damages and causes the irreversible death of auditory cells by apoptosis and necrosis [1,2,4]. These cells do not regenerate in mammals, and therefore, their loss is irreversible and causes hearing impairment. The available solution is prosthesis to substitute for the function of the lost cells, but there are no specific medications [5].

Glucocorticoids are anti-inflammatory and immunosuppressive drugs that have been commonly used as systemic therapy to block the inflammatory substrate in different types of hearing loss [6,7]. Precisely, glucocorticoids are prescribed in high, potentially harmful, doses to alleviate cisplatin-induced hearing loss [8,9]. An alternative local route of administration to reduce side effects could be a transtympanic injection to reach the inner ear through the round window membrane [5]. However, this route poses problems because glucocorticoids are partially washed out from the middle ear, thus reducing drug bioavailability to the inner ear [5]. Additional factors to consider for designing novel smart drug delivery systems (DDS) for glucocorticoids are their low solubility in aqueous media and the fact that their receptors are intracellular [10].

Consequently, there is a medical need to develop DDS to the inner ear to improve local effectiveness and reduce systemic secondary effects. In this context, drug delivery formulations based on hydrogels and nanoparticles have been explored [11,12,13]. Nanoparticles of different chemical natures have been tested for their potential to cross the round window membrane and their capacity to biodegrade [14], including synthetic, amphiphilic, polymeric nanoparticles covalently linked to drugs or interfering RNAs [15,16,17,18]. Lipid nanoparticles are biodegradable and can deliver both hydrophilic and lipophilic drugs depending on their geometry and composition [19,20]. Among the later chemical category, solid lipid nanoparticles (SLNs) are a new generation of submicron-sized lipid particles that present an alternative to the existing lipidic nanocarriers [21]. SLNs have been tested as DDS in a variety of cell types to target different diseases [22,23,24]. SLNs can be loaded with model drugs, such as doxorubicin, paclitaxel or trans-retinoic acid for the treatment of cancer [23,25,26], as well as with hydrocortisone and progesterone, broadly used for the treatment of different diseases [27]. Additional advantages are the well-established production method, the microemulsion, in which organic solvents are avoided, the low cost and easiness with which SLN production can be scaled up [28,29]. Therefore, the use of SLNs offers excellent opportunities for the translation of experimental results to the clinic; however, they have not been tested as DDS for auditory cells.

Here studied whether stearic acid-based SLN with optimal properties of size, pdi and ζ-potential could be nanocarriers of choice for a more effective administration of dexamethasone and hydrocortisone to auditory cells.

## 2. Experimental Section

### 2.1. Materials

Materials Epikuron 200^®^ (containing about 95% soy phosphatidylcholine, PC) was kindly provided by Cargill (Minneapolis, MN, USA). Stearic acid (≥98.5), sodium taurodeoxycholate hydrate (≥95%), hydrocortisone and dexamethasone (Sigma-Aldrich, St. Louis, MO, USA), and octadecyl Rhodamine B chloride (Molecular Probes, Thermo Fisher Scientific, Waltham, MA, USA) were used. Amicon Ultra-15 centrifugal filter units were from Millipore (Darmstadt, Germany) and water purified with a Millipore Milli-Q^®^ ultrapure water unit (Millipore, MA, USA).

### 2.2. Preparation of Solid Lipid Nanoparticles (SLN)

SLN were prepared by the well-established warm microemulsion method, as described in [22,26,27]. Briefly, an oil/water microemulsion was prepared by mixing stearic acid (0.070 mmol), Epikuron 200^®^ (PC) (0.014 mmol) as surfactant, sodium taurodeoxycholate (TDC) as the co-surfactant (0.066 mmol) and ultrapure water as the continuous phase (11.11 mmol), representing molar proportions of 0.62%, 0.12%, 0.59% and 98.66%, respectively. The co-surfactant (TDC) was added in excess to ensure microemulsion formation. Lipids were melted together at 80 °C under continuous stirring (1400 rpm). Water and taurodeoxycholate were then heated to the same temperature. Drugs or fluorescent probes were incorporated into the melted lipid mixture; the aqueous phase was added and the whole mixture stirred until a transparent microemulsion was formed. The microemulsion was then dispersed into cold water (2–4 °C) at a dilution ratio of 1:50 (microemulsion: water, v/v) under vigorous stirring (14,000 rpm for 10 min) with a SilentCrusher M, Heidolph Instruments (Schwabach, Germany), to form an SLN dispersion. Samples were washed three times using Amicon Ultra-15 centrifugal filter units (cut-off 100 kDa, Millipore, MA, USA) to eliminate the excess of TDC [22]. Finally, nanoparticles were suspended in an aqueous sucrose solution at a 15:1 ratio (sucrose: stearic acid, w/w) [30]. We have already showed that the 1:15 ratio provides the minimal amount of sucrose for a proper lyophilisation after testing different sucrose to stearic acid ratios [22]. SLNs were kept at −80 °C overnight, dried under a 50 mTorr atmosphere for 24 h and after lyophilisation were stored at 4 °C until use.

### 2.3. Photon Correlation Spectroscopy

The quality of the SLN used was evaluated by measuring the particle size, polydispersity index (pdi) and ζ-potential by photon correlation spectroscopy. In particular, ζ-potential was measured in distilled water (pH 5.8). Samples were reconstituted in ultrapure water under stirring. Particle size and pdi of SLN dispersions were determined using a Zetasizer Nano S (Malvern Instruments; Malvern, UK). ζ-potential was determined from electrophoretic mobility using a Zetasizer Nano ZS (Malvern Instruments; Malvern, UK). Prior to measurement, each sample was diluted with ultrapure water, as appropriate, and each sample was measured in triplicate at 25 °C. The aforementioned parameters were determined in 5 independently prepared SLN batches [22].

### 2.4. Incorporation of Drugs and Determination of Entrapment Efficiency

SLN samples prepared included: (i) empty SLNs (65 mg SLNs), (ii) SLNs (65 mg) loaded with 20 µg of the fluorescent probe rhodamine (SLN–RHO), (iii) SLNs (65 mg) loaded with 1 mg of dexamethasone (SLN–DEX) and (iv) SLNs (65 mg) loaded with 1 mg of hydrocortisone (SLN–HC). As previously reported [22], drug-containing SLN dispersions were filtered through 0.22 µm pore filters to discard other colloidal populations, such as drug crystals and precipitates that can be formed during abrupt cooling. The incorporation of drugs dexamethasone and hydrocortisone was determined by spectrophotometry [31]. Briefly, SLN suspensions were dried to evaporate the water and dissolved in chloroform, with methanol (1:1 ratio, v/v). Then, the amount of dexamethasone or hydrocortisone incorporated into SLNs was calculated by measuring sample absorbance at 250 nm. Concentration values were obtained using calibration curves built with standard solutions for each drug; measurements were made in duplicate for each sample.

Once the amounts of incorporated drugs were calculated; entrapment efficiency (EE) was calculated applying the following formula:

EE = Amount of drug in SLN/Initial drug amount × 100;

Entrapment efficiency for glucocorticoids was calculated in four SLN–DEX and SLN–HC batches.

### 2.5. Cell Culture Experiments

HEI-OC1 cells were maintained in high-glucose Dulbecco’s modified Eagle’s medium (DMEM, Sigma-Aldrich, St. Louis, MI, USA) supplemented with 10% fetal bovine serum (FBS, Gibco) at 33 °C in a humidified incubator with 10% CO_2_ as described [32].

### 2.6. Confocal Microscopy and Flow Cytometry

To determine SLN–RHO incorporation into HEI-OC1, cells were cultured for 72 h and then incubated with a suspension of SLN–RHO (250 µg/mL) in serum-free medium at different times. The fluorescence intensity of incorporated SLN–RHO was detected using a confocal microscope (LSM710, Carl Zeiss, Jena, Germany). Experiments were performed in triplicate and 3 microphotographs were taken using the same settings for each time point. For flow cytometry experiments, cells were washed and harvested in order to measure SLN–RHO fluorescence using a Cytomics FC500 MPL equipped with MXP software (Beckman Coulter, Indianapolis, IN, USA). Flow cytometry experiments were performed in triplicate.

### 2.7. Cell Cycle Analysis

HEI-OC1 cells were exposed or not to SLN–RHO and samples taken at times 0 and 20 h. Cells were harvested and fixed with 66% ethanol, washed with cold PBS (Phosphate Buffered Saline), suspended in 400 μL of 5 µg/mL 4’,6-diamidino-2-phenylindole (DAPI), plus RNase A (ab139418, Abcam, Cambridge, UK) and incubated at 37 °C in the dark for 30 min. Cell cycle stages were determined using a FACS Canto II cytometer (BD Biosciences, San Jose, CA, USA), and data were processed with BD FACSDiva v.6.1.2 software. Each experimental condition was studied in independent triplicate samples.

### 2.8. Annexin V- Fluorescein isothiocyanate (FITC)/Propidium Iodide (PI) Double Staining and Flow Cytometry

To quantify the percentage of apoptotic cells following SLN–RHO uptake, annexin V-FITC/PI double labeling was used prior to flow cytometry analysis. Briefly, HEI-OC1 cells were collected, washed with PBS and suspended in binding buffer (10 mM HEPES/NaOH (pH 7.4), 140 mM NaCl and 2.5 mM CaCl_2_) (Immunostep, Salamanca, Spain). HEI-OC1 cells were labelled with Annexin V-FITC (Immunostep, Salamanca, Spain) and PI (Abcam, Cambridge, UK) at room temperature for 15 minutes in darkness [33], and then subjected to flow cytometry (Cytomics FC500, Beckman Coulter, Brea, CA, USA) equipped with MXP software (Beckman Coulter, Brea, CA, USA). Each experimental condition was studied in independent triplicates.

### 2.9. In Vitro Cell Cytotoxicity Assay

HEI-OC1 cells were cultured for 72 h and then pre-treated with dexamethasone, hydrocortisone, SLN, SLN–DEX or SLN–HC for an additional period of 24 h. Cell viability was determined using the Cell Proliferation Kit II (XTT assay, 11465015001 ROCHE, Basel, Switzerland), following the manufacturer’s instructions. After reagents addition, cells were incubated for 20 h and colorimetric absorbance was measured at 492 and 690 nm with a VERSAmax microplate reader (VERSAmax tunable microplate reader, Molecular Devices, Sunnyvale, CA, USA). SLN potential toxicity was studied in 2 or 3 independent experiments performed in quadruplicate. SLN–DEX and SLN–HC were tested in 2–4 independent experiments performed in quadruplicate.

### 2.10. Protective Effects of Dexamethasone and Hydrocortisone Loaded-Sln Against Cisplatin

Cells were pre-treated with glucocorticoids (dexamethasone or hydrocortisone) free or incorporated into SLNs (SLN–DEX or SLN–HC) for 24 h; 500 µg/mL loaded SLNs contained either 0.760 µM DEX or 0.240 µM HC, whereas 750 µg/mL loaded SLN contained 1.14 µM DEX or 0.370 µM HC. Then cisplatin (2 or 4 µg/mL, Accord Healthcare Limited, Middlesex, UK) was added to each well. Cell viability was determined 24 and 48 h after cisplatin addition. The biological effects of dexamethasone and hydrocortisone were compared with the effects of SLN–DEX and SLN–HC, respectively, in control (without cisplatin) and ototoxic (with cisplatin) conditions. Cell viability was determined as above.

### 2.11. Statistical Analysis

Data are shown as mean ± standard error of the mean (SEM). All data were analyzed using Sigma Plot (version 11) software (Systat Software, Richmond, CA, USA). The Shapiro-Wilk test was applied to assess normal distribution of the data. Statistical comparison between groups was carried out using analysis of variance (ANOVA) with Dunnet post hoc test. Statistical comparison of means of two data sets was carried out by two-tailed unpaired Students *t*-test. The differences were considered significant for * *p*  <  0.05 and *** *p*  <  0.001.

## 3. Results

### 3.1. SLN Characterization

The quality of SLN, SLN–RHO, SLN–DEX and SLN–HC preparations was evaluated measuring particle size, pdi and ζ-potential values. As shown in Figure 1, all SLNs presented similar values for the three parameters. Therefore, the addition of dexamethasone or hydrocortisone did not alter the characteristics of the nanoparticles. Similarly, incorporation of the fluorescent probe did not modify particle parameters. Specifically, the particle size and pdi values of all SLNs were around 100 nm and 0.3, respectively. Sizes below 150 nm, as shown by these SLNs, are optimal to achieve good uptake rates [21,34], whereas pdi values of 0.3 are considered adequate for drug delivery applications using lipid-based carriers [21]. Regarding ζ-potential, all SLN showed values around −44 mV that ensure the stability of the SLN preparation.

Once SLNs were formed, samples were washed to eliminate the excess of TDC, and 43.4 ± 6 mg of dry HC–SLN and 43.5 ± 7 mg of dry DEX–SLN were recovered. Then drug entrapment efficiency was calculated as described in Section 2.4, with entrapment efficiencies of 21.58% ± 2.08% and 19.42% ± 1.85% for dexamethasone and hydrocortisone, respectively. Taking into account the previous data, drug entrapment efficiencies were equivalent to 9.95 ± 2.98 µg of dexamethasone and 7.55 ± 0.85 µg hydrocortisone per mg of SLNs.

The stability of nanoparticles after lyophilization and storage at 4 °C was evaluated by measuring the above-mentioned parameters after 8 months of storage; all nanoparticles showed unaltered characteristics (Table 1).

### 3.2. Uptake of SLN by HEI-OC1 Otic Cells

The cellular uptake and intracellular localization of SLN–RHO into HEI-OC1 cells were studied by using a combination of confocal microscopy and flow cytometry. Figure 2A shows that SLN–RHO (red) labelled nanoparticles were efficiently internalized by the HEI-OC1 cells showing a punctate distribution in the cytoplasm surrounding the cell nucleus. SLN–RHO uptake was time dependent, showing a significant increase of the uptake efficiency with time (Figure 2A–C). Uptake efficiency of SLN–RHO was quantified by flow cytometry 2, 4, 6, 8 and 24 h after SLN–RHO addition and it increased from 2% to 81% over this period of time (Figure 2C). Therefore, both qualitative (confocal microscopy) and quantitative (flow cytometry) analyses confirmed that SLN–RHO could be efficiently internalized by otic cells.

### 3.3. Cytotoxicity of SLN–RHO and SLN in HEI-OC1 Cells

Next, we studied the effect of SLN–RHO and unloaded SLNs on the HEI-OC1 cell cycle and cell death parameters. Figure 3A–B shows representative plots of cells exposed or not to SLN–RHO (250 µg/mL), in baseline conditions (time 0 h) or incubated for 20 h. No evident interferences in cell cycle progression were observed following exposure to SLN–RHO, and indeed the percentage of cells in G1/G0, S or G2/M cell cycle phases did not vary (Figure 3C). In contrast, the fraction of sub-G0/G1 DNA content slightly but significantly increased upon exposure to SLN–RHO during 20 h (*p* < 0.05) (Figure 3C).

To investigate cell death mechanisms involved, HEI-OC1 cells exposed to SLN–RHO (250 µg/mL) were labelled using the Annexin V-FITC/PI double staining and the percentage of apoptotic cells analyzed by flow cytometry (Figure 3D). Treatment for 20 h with SLN–RHO increased HEI-OC1 cells apoptosis from 4.7% ± 0.1% to 6.2% ± 0.5% of total cells when compared with non-exposed control condition (0 h) (*p* < 0.05 versus control condition) (Figure 3D), confirming that there is a minimal increase of apoptosis upon treatment with RHO loaded SLNs.

The potential cytotoxicity of SLNs for HEI-OC1 cells was further evaluated by incubating cells with increasing concentrations of unloaded SLNs (Figure 3E). HEI-OC1 cell viability was evaluated by using the XTT colorimetric assay; no significant differences were observed between the control (no SLN) and the range of unloaded SLN doses, from 50 µg/mL up to 750 µg/mL, tested. These data indicate that SLN were both efficiently incorporated and biocompatible with the auditory HEI-OC1 cells.

### 3.4. The Effects of Glucocorticoids Alone or Incorporated into SLNs on Hei-Oc1 Cell Viability

Glucocorticoids are the treatment of choice for a variety of hearing loss conditions because they are anti-inflammatory drugs that promote cellular survival [35,36]. Glucocorticoid receptors are intracellular and bind with different affinities to different synthetic molecules, dexamethasone being approximately 27 times more potent than hydrocortisone at the same dose [37,38]. Therefore, next, we made a comparative study between treatments with glucocorticoids alone or loaded into SLNs. HEI-OC1 cells were serum-deprived for 24 h and then treated with increasing doses of dexamethasone or hydrocortisone for 24 h (Figure 4A,B). The dose range was from 0.001 to 100 µM for both glucocorticoids and cell viability was evaluated with the XTT assay. Cells treated with dexamethasone did not show any difference with respect to untreated cells (Figure 4A). Hydrocortisone-treated cells showed a limited but significant increase in cell viability (Figure 4B). Similarly, HEI-OC1 cells were treated with dexamethasone or hydrocortisone loaded into SLNs (SLN–DEX and SLN–HC, respectively) during 24 h. Dose responses of glucocorticoid loaded SLNs, 50 to 750 µg/mL, were carried out, which corresponded to doses in the range 0.076–1.14 µM of dexamethasone (Figure 4C) and 0.025–0.370 µM of hydrocortisone (Figure 4D). Cell viability was not decreased by either SLN–DEX (Figure 4A,C, green bars) or SLN–HC (Figure 4B,D, blue bars) treatments over a wide dose range.

### 3.5. SLN–DEX and SLN–HC Protect Against Cisplatin-Induced Cytotoxicity in HEI-OC1 Cells

Next, we compared the effectiveness of these drugs either alone or loaded into SLN to protect otic cells from cisplatin ototoxicity. Cisplatin decreased the viability of HEI-OC1 cells in a dose and time dependent manner (Figure 5 and Figure 6). The 24 h treatment with 4 µg/mL cisplatin decreased cell viability to 83% (Figure 5A and Figure 6A, grey bars). However, longer treatment times of 48 h decreased cell viability to 65% and 43% with 2 and 4 µg/mL of cisplatin, respectively (Figure 5B and Figure 6B, grey bars).

Glucocorticoids protected HEI-OC1 cells from cisplatin-induced cell death with different time and dose profiles. Dexamethasone (1.14 µM) was able to maintain cell viability under all the conditions tested (Figure 5A,B pale green bars). In contrast, hydrocortisone (0.37 µM) had a protective effect on the cells treated for 24 h with cisplatin 2 µg/mL (Figure 6A, pale blue bars) but was not effective at longer incubation times or against higher doses of cisplatin (Figure 6A,B). Interestingly, SLNs loaded with 0.76 and 1.14 µM dexamethasone, 500 and 750 µg/mL SLN–DEX, respectively, improved protection against cisplatin damage at all the doses and incubation times tested in comparison with free dexamethasone (Figure 5, dark green bars). More strikingly, SLNs loaded with 0.24 and 0.37 µM hydrocortisone, corresponding to 500 and 750 µg/mL SLN–HC, respectively, were able to efficiently protect HEI-OC1 cells from cisplatin-induced apoptosis under all the cytotoxic conditions tested (Figure 6, dark blue bars). Of particular mention is the fact that under the most damaging conditions of treatment with 4 µg/mL cisplatin during 48 h of treatment, doses as low as 0.24 µM hydrocortisone loaded into SLNs showed a better otoprotective effect than free hydrocortisone (0.37 µM). Therefore, SLN–DEX and SLN–HC were more effective formulations for the protection against cisplatin-induced ototoxicity.

## 4. Discussion

Hearing loss has a diverse etiology with genetic and environmental components that, in many cases, have in common, redox unbalance and the consequent inflammation [39,40]. In this context, glucocorticoids have been widely used, particularly dexamethasone to control the inflammatory process and to alleviate the progression of damage [7,41]. Systemic administration of drugs has important side effects, whereas direct administration to the internal ear encounters problems of biodistribution and a short window of temporary action [5,13,42]. Therefore, a great effort is being made to develop novel DDS, and especially, to explore the combination of cochlear implants with new or known drugs that will fulfill the double function of replacing sensory cells and protecting the intracochlear environment [43]. Taking all this into account, nanoparticles have been a preferential object of study in the development of DDS for the treatment of hearing loss. After intratympanic injection, the medication has to maintain sufficient concentration in the middle ear and adequate diffusion through the round window membrane to effectively reach the inner ear. Nanoparticles serve both aspects; loaded-nanoparticles diffuse through the round window membrane and they also facilitate the passage of the freed-drug through it [14]. Nanoparticles containing siRNA have been reported to reach the guinea pig cochlea and recover cellular regeneration by modulating Hes1 [15]. The choice of nanoparticle depends on the chemical properties and speed at which the drug needs to be released, thus lipid nanoparticles are particularly interesting due to their high capacity to incorporate lipophilic molecules, biocompatibility and versatility of routes of administration [20]. Different lipid nanoparticles have been tested as delivery systems to the inner ear. For instance, phospholipid-based nanoparticles containing liquid seed oil core have rendered positive results [44]. However, nanoparticles containing a solid core have been reported to be a better option than liquid systems for controlled release of the drugs [45,46,47,48]. In this context, the characteristics of SLN nanocarriers that are formed by biocompatible and biodegradable lipids, which are solid at body temperature, seem optimal for controlled drug-delivery [49]. However, despite their potential, only Gao and co-workers [50] have described the use of SLNs to treat noise-induced hearing loss. The work presented here aimed to characterize SLNs as nanocarriers to deliver drugs to the inner ear, and concretely, to study whether stearic acid-based SLNs could facilitate the use of glucocorticoids as local co-therapy with systemic cisplatin to prevent ototoxicity.

Stearic acid-based SLN characterization showed that particle size (≈100 nm), pdi (≈0.3) and ζ-potential (≈−44 mV) values were compatible with optimal DDS [21]. All SLNs generated in this work, SLN, SLN–RHO, SLN–DEX and SLN–HC, presented very similar physical parameters, indicating that glucocorticoid incorporation did not alter SLN characteristics. Therefore, the results obtained with unloaded and glucocorticoid-loaded SLNs are comparable. Particle size is of key importance to maintain the desired balance between clearance of the drug from the middle ear cavity and drug permeation into the cochlea. Generally, nanoparticles in the range 100–200 nm are reported to be well taken up by a variety of cell types, and avoid elimination by clearance systems [51]. In the ear, it has been reported that nanoparticles smaller than 200 nm efficiently go through the round window membrane [52]. However, it is worth mentioning here that there is no unanimous vision on this point as other reports suggest that even nanoparticles >200 nm are efficiently transported into the cochlea [53]. Accordingly, Yang and co-workers [44] studied four types of phospholipid-based nanoparticles with sizes greater than 200 nm and concluded that the best drug delivery was carried out by cationic-PEG carriers of size 278 nm. As most previous studies in inner ear drug delivery used particles in the range 50–170 nm [17,50,54,55], SLNs tested here had particle sizes of around 100 nm.

Pdi indicates the quality of size distribution of a nanoparticle preparation [21]; pdi values ≤0.1 are considered of high quality of dispersion, whereas values ≤0.3 are optimal [49]. Accordingly, in the inner ear, efficient delivery has been reported with nanoparticles of similar pdi values [16,53]. The SLNs studied here had pdi values around 0.3, thus optimal for drug delivery to auditory cells. Finally, ζ-potential indicates the stability of the colloidal dispersion because charged particles of either high positive or negative ζ-potential are less prone to aggregate due to electrostatic repulsions, and are, therefore, more stable [56]. Other nanoparticles used in inner ear therapy showed lower ζ-potential values and limited efficacy [44,53,54,57,58]. We prepared SLNs with highly negative ζ-potentials (≈ −40 mV), indicative of their high stability. From the pharmacological point of view, the possibility of generating batches of SLNs that can be stored assures the reproducibility of the treatment. Previous results from our group showed that the presence of sucrose improved the stability of SLNs when lyophilized [22]. DDS reported previously showed stability over short storage times [53,58]. Here, we show that after 8 months of storage at 4 °C, particle size, pdi and ζ-potential values of lyophilized SLN persisted unaltered. This is one of the few studies in which long-term storage has been reported and reinforces that stearic acid based SLNs are candidates to develop pharmaceutical formulations.

Several drugs have been incorporated into nanoparticles to study their otoprotective capacity. Here we have used glucocorticoids commonly used as co-therapies in anti-cancer treatments with cisplatin to reduce hearing loss without altering its antimitogenic potential. Here we report that SLN loaded with either dexamethasone or hydrocortisone were efficiently up-taken by auditory cells (2–24 h) and that the glucocorticoid effective dose lowered with respect to those previously reported by using other nanocarriers [17,18,27,44,55,58]. Notably, SLNs were not cytotoxic and no differences in cell viability were observed in the range tested, 50–750 µg/mL. SLN–RHO showed a negligible 1.33% increase at the highest dose tested. These values are well below or in line with those previously reported for other nanoparticles delivered to otic cells [17,44,58], although the disparity of the experimental models used does not allow further comparisons. We have studied here in detail SLN incorporation into auditory HEI-OC1 cells, a well-established model for the study of ototoxicity and drug screening [59].

As mentioned earlier, glucocorticoids are used for their anti-inflammatory activity in the treatment of various types of sensorineural hearing loss, including being secondary to antitumor treatment with cisplatin. Glucocorticoids have intracellular receptors [10] and the mechanisms of action involved in otoprotection have been extensively studied [60] Glucocorticoids have been prescribed using different routes of administration and posology, in the clinic and in experimental models to develop more effective treatments of hearing loss [61,62]. Dexamethasone is a long-acting glucocorticoid with an anti-inflammatory potential about 25 times higher than that of short-acting products, such as hydrocortisone. Dexamethasone’s effect is reached between 24 and 36 h and it is indicated for chronic anti-inflammatory treatments. In contrast, hydrocortisone is used in acute anti-inflammatory treatments because it acts more quickly, within minutes, and has fewer side effects [63,64]. Increased glucocorticoid dosage or duration of therapy increases the risk of undesirable effects. Whenever possible, the goal of steroid therapy should be to maintain the lowest dosage that results in adequate clinical response. Therefore, loading of glucocorticoids in SLNs allows for a topical application; looking for local effects facilitates a slower delivery and contributes to the target of decreasing dosage. In the case of hydrocortisone, SLN nanocarriers prolong its bioavailability, thus its activity window, which greatly improves the response profile over long time periods. The HEI-OC1 cell line has also been extensively used to study the response to cisplatin and otoprotection by different drugs, including glucocorticoids [59]. Here we show that glucocorticoids loaded into SLNs are more efficient in protecting from cisplatin-induced HEI-OC1 apoptosis than free glucocorticoids at the same dose. The potential that the preparation in SLNs give to hydrocortisone is especially striking; it exceeds dexamethasone in the level of protection achieved against the highest damaging cisplatin condition tested. Dexamethasone, but not hydrocortisone, preparations in different nanoparticles have been reported, but the otoprotection achieved here is superior [55,65,66].

In conclusion, here we show that: (i) stearic acid-based SLNs present optimal characteristics to be used as DDS, including low size, polydispersity index and high ζ-potential; (ii) HEI-OC1 cells uptake SLNs efficiently, and we have established a working window at which these nanoparticles are safe; and (iii) SLNs loaded with dexamethasone and hydrocortisone exhibited higher otoprotection against cisplatin in terms of increased effectiveness than the same drugs alone. Our results taken together suggest that drug encapsulation into SLNs can be advantageous in terms of vehiculization and effectiveness for hearing loss protection.

## Figures and Tables

**Figure 1 jcm-08-01464-f001:**
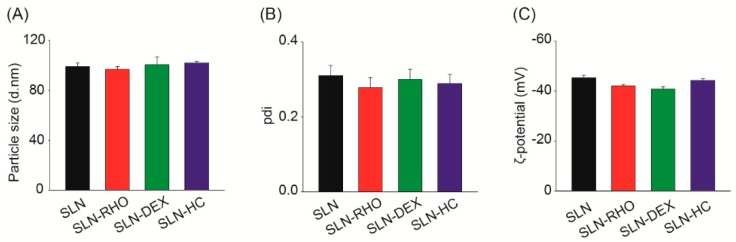
Particle size, polydispersity index and ζ-potential of solid lipid nanoparticles (SLNs). (**A**) Particle size, (**B**) polydispersity index (pdi) and (**C**) ζ-potential values of different SLN suspensions: empty SLNs (SLN), SLNs loaded with rhodamine (SLN–RHO), SLNs loaded with dexamethasone (SLN–DEX) and SLNs loaded with hydrocortisone (SLN–HC). Data are shown as the mean ± SEM from five independent experiments.

**Figure 2 jcm-08-01464-f002:**
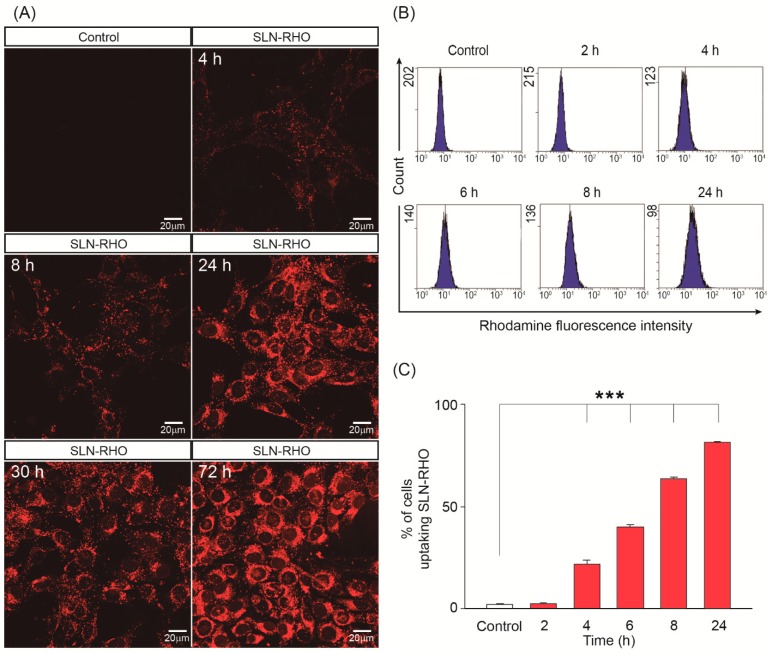
Uptake of SLN–RHO into HEI-OC1 cells. (**A**) HEI-OC1 cells were incubated with 250 µg/mL of rhodamine-loaded SLN (SLN–RHO). Fluorescence intensity of incorporated SLN–RHO was detected by confocal microscopy at the times indicated. Representative microphotographs from three independent experiments are shown. (**B**) HEI-OC1 cells were incubated with SLN–RHO (250 µg/mL) for different times, cells were then collected, and the fluorescent intensity of incorporated SLN–RHO was detected by flow cytometry. The plots shown are representative of three independent experiments, whose quantification is shown in (**C**). Data are shown as the mean ± SEM from three independent experiments. One-way ANOVA was used to determine statistical significance; ****p* < 0.001.

**Figure 3 jcm-08-01464-f003:**
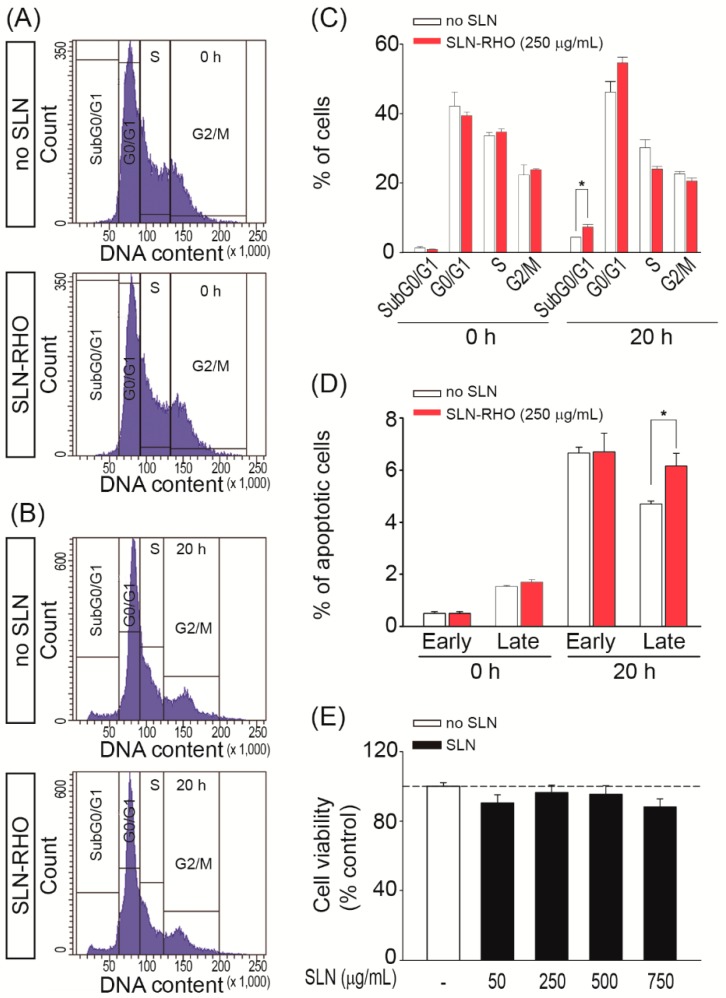
The cytotoxicity of SLN–RHO and SLN in HEI-OC1 cells. (**A**) Flow cytometry analysis of cell cycle phase distribution in HEI-OC1 cells. Representative profiles of DAPI staining from control (no SLN) and SLN–RHO (250 µg/mL) treated cell cultures, at 0 h (**A**) and 20 h (**B**). (**C**) Histogram showing the cell distribution in the sub-G0/G1, G0/G1, S and G2/M phases in control and SLN–RHO (250 µg/mL) treated-cells. (**D**) HEI-OC1 cells were incubated or not with SLN–RHO for 20 h, stained with FITC-conjugated Annexin V and PI, and the pencentages of cell populations in early and late apoptosis were quantified. Data are shown as the mean ± SEM from three independent experiments. The significance of the differences was evaluated using Student´ *t* tests; **p* < 0.05 versus control condition (without SLNs). (**E**) HEI-OC1 cells were seeded into 96-well microplates 1500 cells/well and treated with the doses indicated of SLN for 24 h. Cell viability was determined with the Cell Proliferation Kit II (XTT) assay. Results are the mean ± SEM of two-three independent experiments performed in quadruplicate. The significance of the differences was evaluated using one-way ANOVA testing; * *p*  <  0.05.

**Figure 4 jcm-08-01464-f004:**
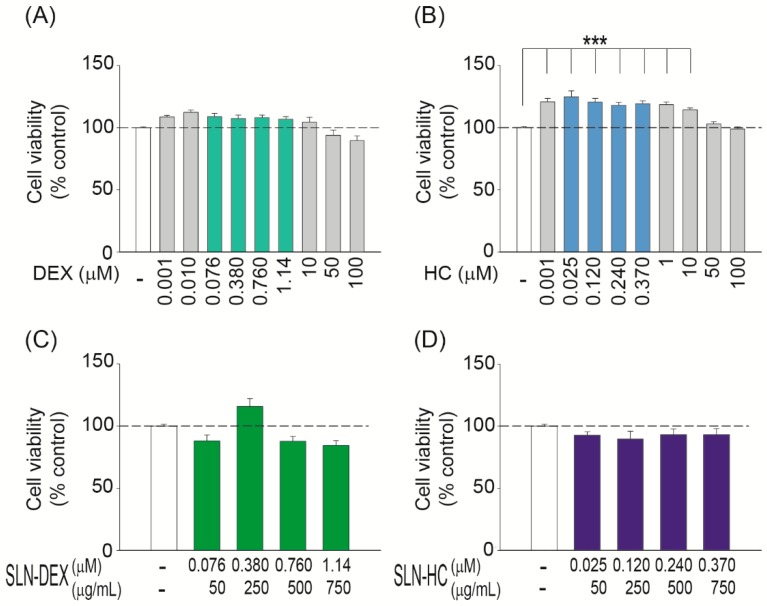
Differential effects of free versus SLN-incorporated glucocorticoids on HEI-OC1 viability. Cell viability of HEI-OC1 treated with different concentrations of dexamethasone ((**A**), grey or pale green), hydrocortisone ((**B**), grey or pale blue), SLN–DEX ((**C**), dark green) or SLN–HC ((**D**), dark blue) over 24 h. The a and b green and blue bars also correspond to the doses tested for dexamethasone or hydrocortisone incorporated into SLN in (**C**) and (**D**), respectively. Cell viability was analyzed using the XTT assay as described in the Methods section. Results are shown as mean ± SEM. (**A**–**B**) data were obtained from two independent experiments performed in quadruplicate, whereas (**C**–**D**) data were obtained from two-four independent experiments performed in quadruplicate. The significance of the differences was evaluated using one-way ANOVA testing; ****p* < 0.001.

**Figure 5 jcm-08-01464-f005:**
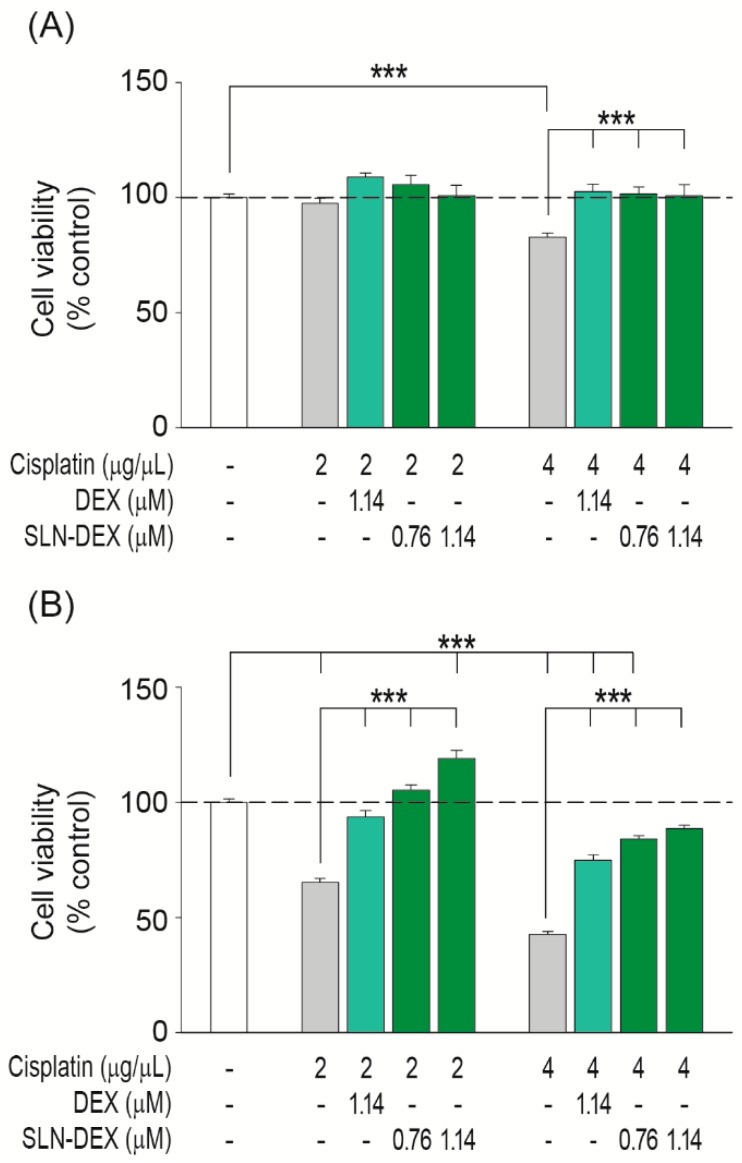
Comparative effects of SLN–DEX and dexamethasone on HEI-OC1 cells’ viability. Cell viability was evaluated by Cell Proliferation Kit II (XTT) assay. HEI-OC1 cells were treated for 24 h (**A**) or 48 h (**B**) with different concentrations of dexamethasone (DEX) either alone or combined with SLNs (SLN–DEX) in the absence or presence of 2 or 4 µg/mL cisplatin. Results are shown as mean ± SEM from four independent experiments. The significance of the differences was evaluated using one-way ANOVA testing; ****p* < 0.001.

**Figure 6 jcm-08-01464-f006:**
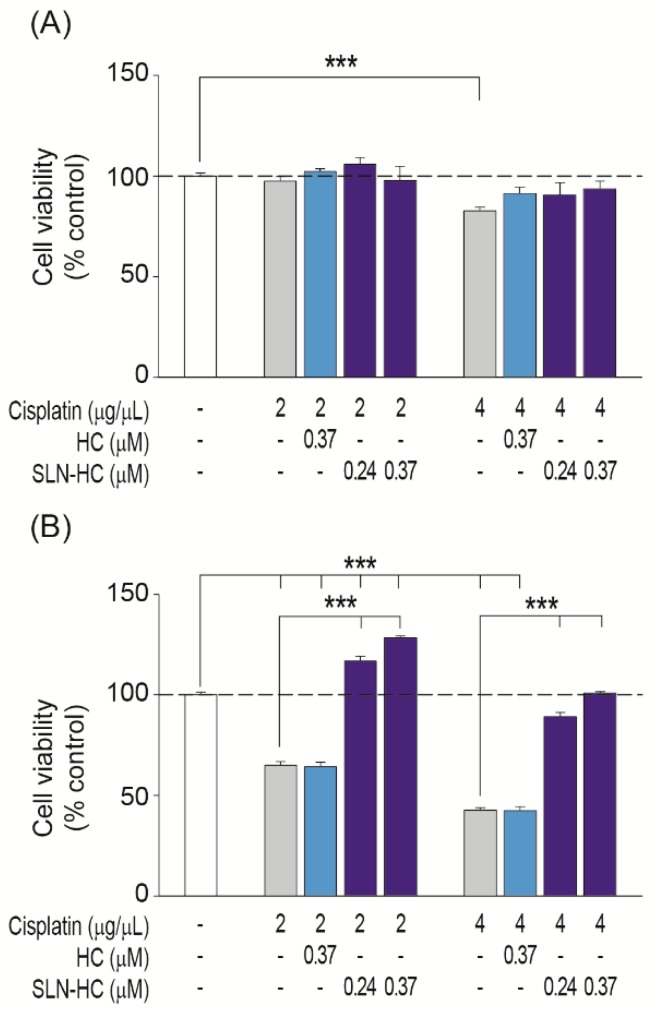
Comparative effects of SLN–HC and hydrocortisone on HEI-OC1 cell viability. Cell viability was evaluated by Cell Proliferation kit II (XTT) assay. HEI-OC1 cells were treated for 24 h (**A**) or 48 h (**B**) with different concentrations of hydrocortisone (HC) either alone or combined with SLNs (SLN–HC) in the absence or presence of 2 or 4 µg/mL cisplatin. Results are shown as mean ± SEM from four independent experiments. The significance of the differences was evaluated using one-way ANOVA testing, ****p* < 0.001.

**Table 1 jcm-08-01464-t001:** Characterization of SLNs with different surface chemistries.

Nanoparticle		Particle Size ± SD (nm)		pdi ± SD		ζ-potential ± SD (mV)	
	Storage (Month)	0	8	0	8	0	8
SLN		99.00 ± 4.96	94.03 ± 6.13	0.309 ± 0.05	0.333 ± 0.031	−45.3 ± 1.82	−41.81 ± 1.88
SLN–DEX		100.69 ± 10.23	117.98 ± 5.70	0.299 ± 0.03	0.278 ± 0.009	−44.3 ± 1.09	−46.00 ± 2.83
SLN–HC		101.91 ± 2.19	124.33 ± 9.01	0.288 ± 0.05	0.232 ± 0.016	−44.8 ± 1.58	−44.38 ± 3.48

Properties of SLN, SLN–DEX and SLN–HC, particle size, pdi and and ζ-potential, were measured before and after 8 months of storage (*n* = 2).

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
