# Peer review of "Solid Lipid Nanoparticles Loaded with Glucocorticoids Protect Auditory Cells from Cisplatin-Induced Ototoxicity"

_jcm, 2019, doi:10.3390/jcm8091464_

Round 1

Reviewer 1 Report

The authors examined nontoxicity and efficacy of solid lipid nanoparticles (SLNs) loaded with glucocorticoids in HEI-OC1 immortalized auditory cells. They showed that the stearic acid-based SLNs has optimal characteristics as a drug delivery system, including low size, polydispersity index and electrostatic stability. SLNs were uptaken by HEI-OC1 cells with minimum cytotoxicity. SLNs loaded with dexamethasone and hydrocortisone exerted higher otoprotection against cisplatin than the glucocorticoids alone.

The manuscript is well written and provides reliable data on the topic, as long as it is conceived that the present data is confined to SLN uptake by in vitro HEI-OC1 cells.

Major comments

 1)    As a future perspective for the SLN administration in vivo, the authors discuss about possible intratympanic administration and bioavailability of SLN through the round window membrane. Please mention how the authors think about systemic administration of SLN (e.g. intravenous administration) in vivo. Do the authors think SLNs pass through the blood-labyrinthine barrier in the inner ear?  How do the authors expect about the systemic toxic effect of SLNs given SLNs were administered systemically?

2) As shown in figures 5 and 6, SLN-dexamethasone and SLN-hydrocortisone exerted more effective otoprotection against cisplatin than dexamethasone and hydrocortisone alone. These figures are most important data that represent the ideas described in this manuscript.

The difference between the otoprotection by SLN-hydrocortisone and hydrocortisone alone is more drastic that that between SLN-dexamethasone and dexamethasone alone. Please mention more explicitly why SLN-hydrocortisone showed more distinct superiority than hydrocortisone alone, than SLN-dexamethasone as compared to dexamethasone alone.

3) Similarly, please discuss about intracellular molecular mechanisms by which glucocorticoids protect against cisplatin ototoxicity. The below reference may provide useful information.

Dexamethasone Protects Against Apoptotic Cell Death of Cisplatin-exposed Auditory Hair Cells In Vitro.Dinh CT, Chen S, Bas E, Dinh J, Goncalves S, Telischi F, Angeli S, Eshraghi AA, Van De Water T.Otol Neurotol. 2015 Sep;36(9):1566-71.

Minor comment.

Line 183, “Values of p<0.001 were considered statistically significant.”

Is this adjusted p-value for the post-hoc test? It is unclear for this reviewer why this relatively stringent significance level was used. The significance level for t-test is p<0.05.

Reviewer 2 Report

The paper presents interesting data on the efficacy of a drug delivery system (DDS) based on solid lipid nanoparticles against cisplatin toxicity in an in vitro inner ear cell model. The paper is well structured and written, and the topic is important in the field of inner ear therapy. I appreciated the comparison between the two anti-inflammatory drugs and the comparison of the effect of the drugs applied with and without DDS. Besides the good results achieved, the authors should improve the paper with minor revision concerning the references, the statistical analyses, the material method and data description before publication.

Specific comments:

Page 2, line 46, 50, 51, 54. Add references.

Page 2, line 63. The references 15 and 16 did not concern inner ear.

Page 2, lines 76-81. The paragraph anticipate the paper conclusion. In my opinion  is better to avoid this anticipation in the introduction, while it is essential to highlight the importance of, and the need of the present study.

Page 3, paragraph 2.4. Please could you report how much SLN were loaded with the three compounds (RHO, DEX, HC)?

Page 4, paragraph 2.11. Please could you specify if it was tested the normality of the data distribution before the application of the parametric analyses  ANOVA and t-test?

Page 5, lines 197-200.  Sorry, it is not clear how did you measure the amount of DEX or HC entrapped, the author should rewrite this paragraph.

IN THE RESULTS SECTION

Why did you show the uptake of SLN –RHO and the cell cycle of SLN at the dose of 250ug/ml and then you studied the effect of SLN DEX and SLN HC at 500 and 750 ug/ml?

Page 7 Figure 3 A) and B) which graph refer to control and which one to SLN-RHO?

Page 6 why did you report the SLN uptake before the SLN cytotoxicity?

Page 8 line 280. “Cell viability was not modify”: comparing figure 4B to 4D it doesn’t look unmodified.

Figure 9 and 10. Why did you not inserted the data related to HEI-OC1 treated with respectively 0.76 uM DEX and 0.24 uM HC   
